# Effect of Different Sowing Methods on Water Use Efficiency and Grain Yield of Wheat in the Loess Plateau, China

Hafeez Noor [1,2] , Min Sun [1,2,*], Wen Lin [1,2] and Zhiqiang Gao [1,2]

1   College of Agriculture, Shanxi Agriculture University, No.1, Minxian South Road, Taiyuan 030006, China; hafeeznoorbaloch@gmail.com (H.N.); slwrdewy@163.com (W.L.); gaosxau@163.com (Z.G.)
2   State Key Laboratory of Sustainable Dryland Agriculture (In Preparation), Shanxi Agricultural University, Taiyuan 030006, China
*   Correspondence: sm_sunmin@126.com

**Abstract:** Research has revealed that summer fallow sowing improves the water use efficiency (WUE) and grain yield of winter wheat. However, wheat yields differ yearly with crop management. A field experiment over 8 years was established in the Loess Plateau to determine the role of precipitation and soil water storage in wheat yield formation under conservation tillage. The average WUE values were 7.8, 11.0, and 12.6 t·ha$^{-1}$, while the average evapotranspiration (ET) values were 334.7, 365.5, and 410 mm when the yields were 3.0, 3.0–4.5, and over 4.5 t·ha$^{-1}$, respectively. Compared to drill sowing (DS), high water consumption during early growth increased the spike number, grain number, and yield. In years of intermediate or low yields, wide-space sowing (WS) and furrow sowing (FS) improved the ET, WUE, spike number, grain number, and yield of wheat compared to (DS) drill sowing. When the wheat yield was 3.0–4.5 t·ha$^{-1}$, higher soil water intake during jointing, anthesis, and anthesis–maturity increased the tiller number, 1000-grain weight, and yield, related to the use of suitable tillers. Synchronous increases in grain number per spike and 1000-grain weight were observed with increased soil water content at jointing, maturity, and anthesis, as well as consumption of soil water in the latter part during the growing season.

**Keywords:** evapotranspiration; precipitation; soil water storage; water; wheat yield level

## 1. Introduction

The Loess Plateau is the dominant region for cereal crop production in China. In this region, wheat (*Triticum aestivum* L.) covers about 56% of the arable land [1], restricted by the extraordinary variability in precipitation and evaporation during the summer fallow period [2,3]. Agriculture has been exploited in this area to guarantee food security, which has accelerated ecological deterioration, including soil physical structure degradation, water and soil pollution, and reduced crop productivity [4–6].

The production of winter wheat in dryland is important for regional food security [7,8]. In the Loess Plateau dryland, irrigation is not available, and rainfall is the only source of water for the production of wheat. Precipitation levels are low and unevenly distributed, whereby summer rainfall accounts for approximately 60% of the annual precipitation [9,10]. Furthermore, annual precipitation fluctuates considerably [11]. Because of the limited water resources, the main planting approach in this area is to plant one crop (winter wheat) per year and leave the land fallow in the summer [12,13].

Many agricultural management plans have been established in the past few years to improve crop production in dryland regions, with one of the most successful methods being conservation tillage, with permanent organic soil cover and mechanical soil disturbance [7]. Conservation tillage approaches include DS (drill sowing), FS (furrow sowing), and WS (wide-space sowing), and they play an active role in increasing crop yield. FS, which usually includes straw mulching, leads to reduced soil degradation and farmland erosion caused by intensive agriculture [14,15]. Previous studies reported the adverse effects of

FS on soil properties such as improved soil bulk density and reduced total porosity and penetration resistance. Additionally, "reduced tillage" practices, such as WS, whereby soil is usually chisel-plowed to a depth of 25 cm, or DS, in which case soil is frequently chisel-plowed to a complexity of 20 cm, are used to alleviate soil compaction by breaking hardpans [16,17].

These reduced sowing practices have a positive outcome on rain penetration into the soil and water storage, thus improving the soil water content and increasing the tiller number, wheat grain yield, and plant WUE. The excessive use of nitrogen (N) fertilizers can have several negative effects on the environment [18]. A previous study showed that the use of controlled-release nitrogen fertilizers at sowing increased the crop yield, WUE, and economic returns by 8.5%, 10.9%, and 11.3%, respectively [19]. Another study on N fertilizers in the Loess Plateau reported that the application of an appropriate amount of N fertilizer increased the content of total wheat protein and composition of protein, leading to an improvement in the baking quality of wheat flour [20].

With an increase in the N application rate, the investment in N application should be determined to optimize economic return [21]. Soil fertility in the Loess Plateau dryland is low, especially the N level [22]. The application of N fertilizers can significantly increase the grain yield and WUE of wheat [23]. Wheat yield components include the number of tillers, grain number per spike, and 1000-grain weight. Increased coordination among yield components is required to improve crop yield potential [12–14]. However, some studies have shown that the contribution of the various yield components differs with the yield level, and correlation analysis between any single variable and yield does not fully explain the importance of each component [23–25].

Furthermore, some studies have shown a significant correlation influenced by field water consumption between wheat yield and soil moisture status over multiple growth stages from sowing to maturity [26,27]. Apparently, the sowing methods applied by farmers in the area exceed the level of sowing required to achieve high yield. The effects of water on yield, as well as the response of yield to sowing, vary with the annual precipitation level. A study on different sowing approaches in the Loess Plateau for eight consecutive years showed that, when 150 kg N·ha$^{-1}$ was applied, wheat yield in the dry years increased by 14.0% relative to no nitrogen application, whereas it increased by 32.8% in the wet years [18] Therefore, the optimization of sowing methods based on precipitation is important to achieve high wheat yield in dryland while improving grain quality, economic return, and WUE (water use efficiency).

In this study, the main objectives were to determine the correlation between yield and soil water content and water consumption at different yield levels, thereby allowing (1) clarification of the correlation between soil water content at different yield levels and plant growth stages, (2) a comparison of the differences in yield components and WUE, and (3) an evaluation of the relationship between grain yield or yield components and field water consumption during key plant growth stages.

## 2. Materials and Methods

During the winter, field experiments were conducted for the winter wheat growing seasons in the years 2009–2017 at the experimental station of the Shanxi Agricultural Wenxi region, China. The study area was located in the Wenxi region (34°35′ N; 110°15′ E), Shanxi province, in the southeast of the Loess Plateau is shown in Figure 1.

This region is characterized by a distinctive semiarid, warm temperate continental climate with an annual mean precipitation of 491 mm, annual mean temperature of 12.9 °C, annual sunshine period of 2242 h, and open pan evaporation of 1839 mm. Although the annual precipitation tends to be concentrated in the months of July through September, it displays great annual variability. The precipitation distribution over the years 1981–2017 is shown in Figure 2.

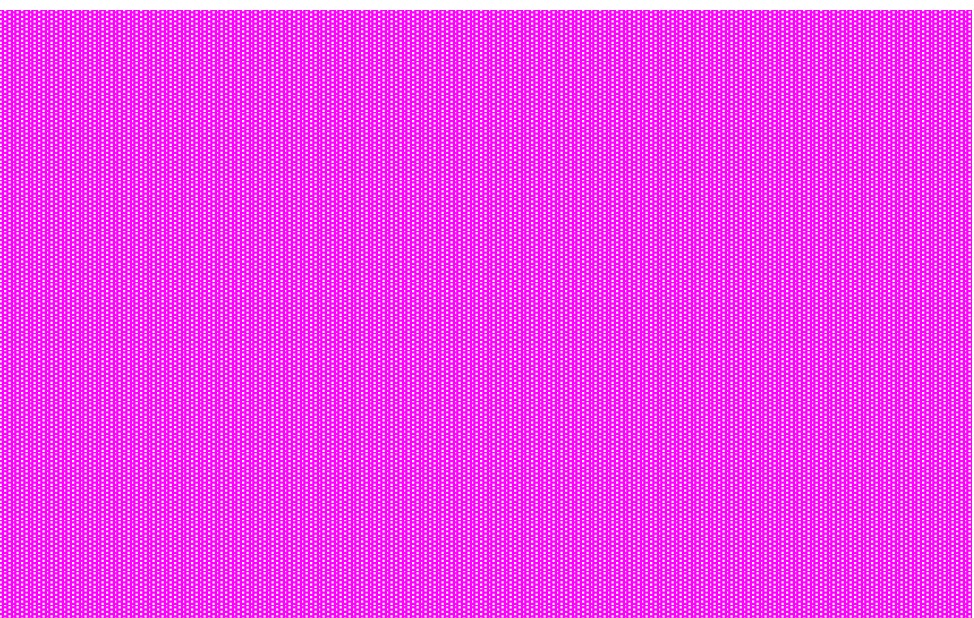

**Figure 1.** Location of experiment site in the Loess Plateau. The regional distribution of annual precipitation is shown in different colors on the map.

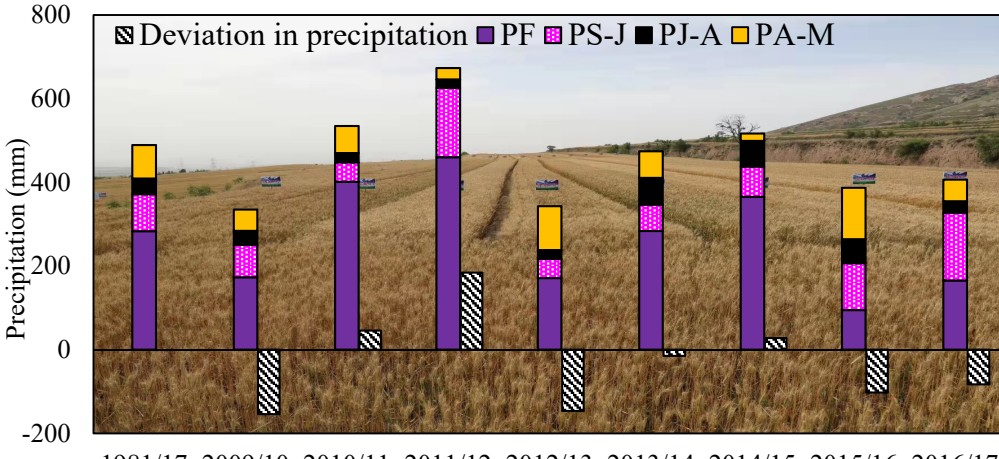

**Figure 2.** Precipitation distribution in the study area from 2009 to 2017, during the growth stage of winter wheat and the fallow season. $P_F$, $P_S$-J, $P_J$-A, and $P_A$-M denote the precipitation during the fallow, sowing, anthesis, and maturity stages of wheat, respectively.

## 2.1. Experimental Design and Field Management

This experiment featured a single-factor randomized block design. Winter wheat (*Triticum aestivum* L.) cultivar 'Yunhan 20410' was acquired from the Shanxi Agriculture Bureau, Wenxi, China. The trial comprised three different sowing methods: (1) wide-space sowing (WS) (sowing spacing and row spacing of 8 and 25 cm, 2BMF-12/6, with auto-fertilization and tillage), (2) furrow sowing (FS) (ridge height 3/4 cm, furrow depth 6/7 cm, narrow and wide spacing 10/12 cm and 20/25 cm, and 2BMFD-17/14 multi-resolution), and (3) drill sowing (sowing spacing and row spacing of 3 cm and 20 cm, 2BXF-12 seed drill) (Figure 3).

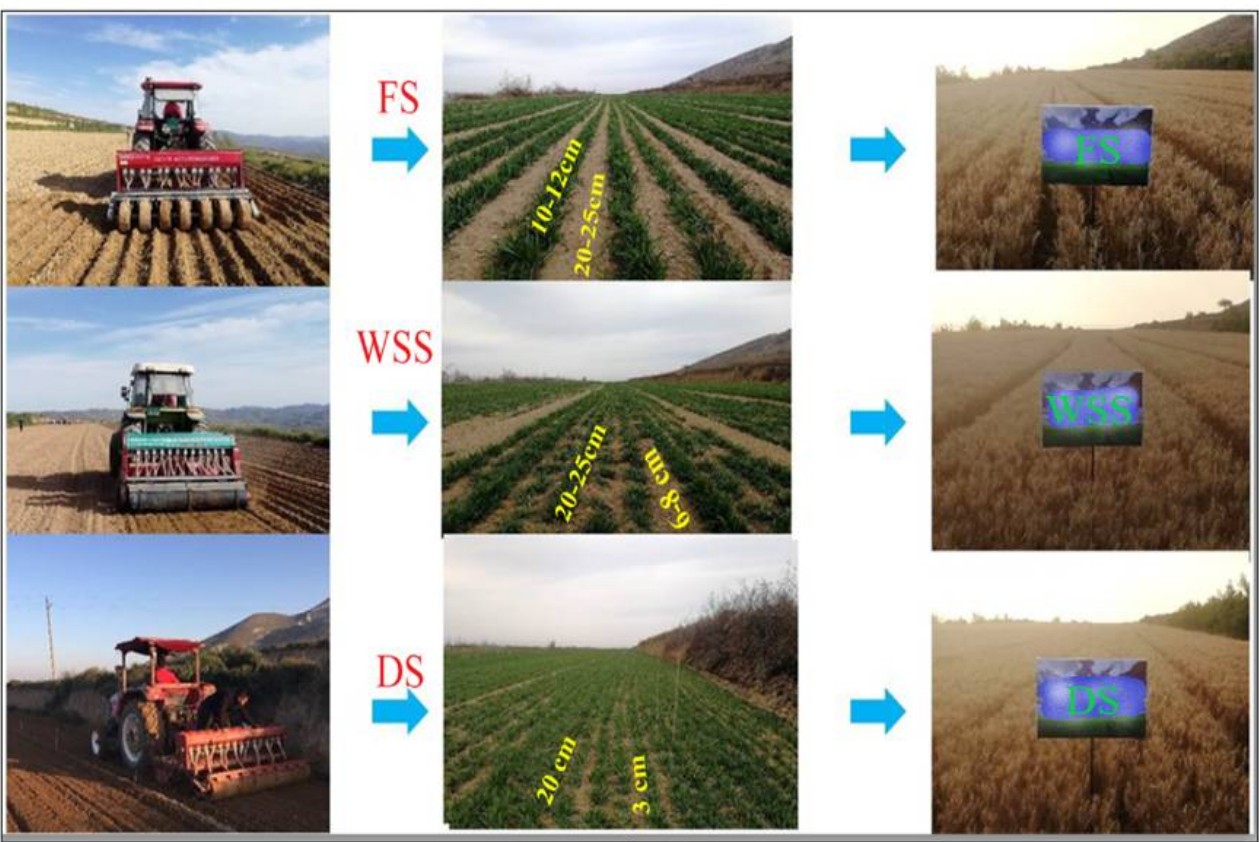

**Figure 3.** Illustration of sowing methods with row spacing (FS, furrow sowing; WS, wide-space sowing; DS, drill sowing), at different growth stages of wheat in the study area, Shanxi Wenxi, China.

Each plot had an area of 300 m$^2$ (6 m × 50 m). Before planting, 150 kg N·ha$^{-1}$ (urea 46%), P$_2$O$_5$ (38 kg·ha$^{-1}$), and K$_2$O (75 kg·ha$^{-1}$) were applied consistently to the soil is shown in Table 1. During each cropping season, the planting density was 315 × 10$^4$ plants·ha$^{-1}$. During each year, all plants were machine-harvested in late June. Throughout the growing season, weeds were manually controlled, and no irrigation was applied at any time during the entire experimental period.

**Table 1.** Basic soil properties of the 0–20 cm layer in the study area from 2012–2017.

| Year | Organic Matter (g·kg$^{-1}$) | Total N (g·kg$^{-1}$) | Alkali-Hydrolysis N (mg·kg$^{-1}$) | Available Phosphorus (mg·kg$^{-1}$) |
|---|---|---|---|---|
| 2012–2013 | 8.63 | 0.71 | 32.89 | 15.73 |
| 2013–2014 | 9.18 | 0.70 | 39.32 | 16.62 |
| 2014–2015 | 9.55 | 0.68 | 37.65 | 17.64 |
| 2015–2016 | 8.54 | 0.67 | 32.79 | 19.23 |
| 2016–2017 | 9.62 | 0.69 | 32.22 | 15.28 |
| 2017–2018 | 8.07 | 0.69 | 33.42 | 16.26 |

*2.2. Measurements*

2.2.1. Soil Moisture

Soil water storage (SWS, mm) and soil gravimetric moisture content (GSW%) were measured gravimetrically at each plant growth stage. Soil samples were collected from a depth of 300 cm at 20 cm intervals [28]. One sample was measured as one replicate. GSW and SWS were obtained using Equations (1) and (2), respectively.

$$\text{GSW}(\%) = \frac{\text{Mw Md}}{\text{Md}} \times 100, \tag{1}$$

$$SWS \ (mm) = GSW \ (\%) \times \rho b \left(g \ cm^{-3}\right) \times SD \ (cm), \tag{2}$$

where $M_w$ and $M_d$ are the weights (g) of dry and wet soil, respectively, pb is the soil bulk density of the given soil layer, and SD is the soil depth.

2.2.2. Evapotranspiration (ET), Precipitation, and Water Use Efficiency (WUE)

Precipitation (mm) and consumption of stored soil water (mm) in the 0–300 cm layer were used to calculate the c WUE, PUE, and evapotranspiration (ET) rate for a given cropping season using Equations (3)–(5).

$$ET = SW_0 - SW_1 + P - R - D, \tag{3}$$

$$WUE \left(kg \cdot ha^{-1} \cdot mm^{-1}\right) = grain \frac{yield}{ET}, \tag{4}$$

$$PUE \left(kg \cdot ha^{-1} \cdot mm^{-1}\right) = grain \frac{yield}{P}, \tag{5}$$

where $SW_0$ is the soil water storage before sowing, and $SW_1$ is the soil water storage after harvest. P is the precipitation during the wheat growth period, R is the soil surface runoff, D is the deep percolation, and Pt is the total precipitation from tillage to harvest. The experimental field was flat, and the experimental plots were surrounded by ridges to inhibit runoff; in this research, R was estimated to be 0. The ground water table was deeper than 50 m in the study area, and no water was percolated to the deep soil layers; therefore, D was also considered to be 0.

Precipitation (mm) and intake of soil water for storage (mm) in the 0–300 cm layer were used to calculate the crop water consumption during different growth periods. The sum of precipitation (mm) and intake of soil water for storage (mm) in the 0–300 cm layer from sowing to plant maturity was taken as the evapotranspiration (ET) rate for a given cropping season.

2.2.3. Yield and Yield Components

Fifty plants per plot were randomly sampled at maturity from the inner rows to determine yield components including ear number and grain number per ear. Plot grain yield was determined by harvesting all plants in an area of 20 $m^2$ and shelling them mechanically. Then, the grain was air-dried for determination of grain yield.

2.3. Statistical Analysis

The data of winter wheat growth and yield formation were processed and statistically analyzed using SAS-8.6 (SAS Institute Inc., Cary, NC, USA). In this study, two-way ANOVA was used to determine the main soil water storage and types of yield formation. When there was a significant interaction effect between soil water and yield, the least significant difference (LSD) method was used for differential analysis, while the F-test was used to determine independence; the significance level was set to $\alpha = 0.05$. Differences were considered statistically significant when $p \leq 0.05$.

3. Results

3.1. Soil Water Storage

The association between yield development and soil water storage fluctuated with the yield level (Figure 4). Yield was not significantly related to soil water storage at the jointing or anthesis stages; however, with increasing soil water storage, yield first decreased and then increased. This indicated that soil water storage was higher than 388.2 mm, 331.2 mm, and 258.0 mm at the sowing, jointing, and anthesis stages, respectively (Figure 4A–C). At the intermediate yield level, yields increased with soil water storage, with the maximum soil water storage at the jointing stage (Figure 4B). Lastly, at a high yield level, yields were mostly correlated with soil water storage at the jointing, anthesis, and maturity stages.

This trend was similar to that observed for the intermediate yield level (Figure 4A–C). Our results indicate that higher soil water storage during the late stages of growth is crucial for a higher yield.

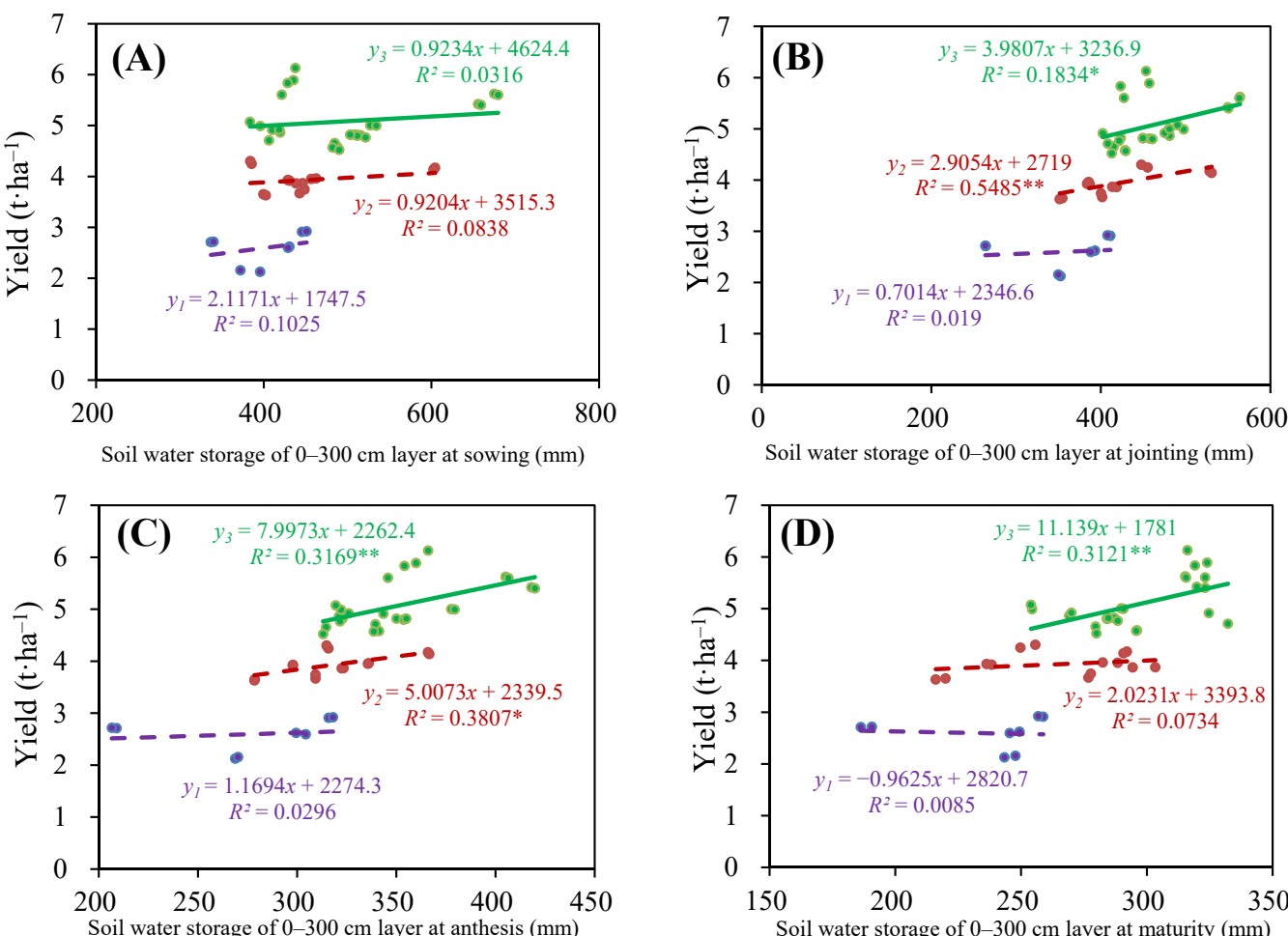

**Figure 4.** Correlation between soil water storage and sowing stage ($y_2$ = FS, furrow sowing; $y_3$ = WS, wide-space sowing; $y_1$ = DS, drill sowing); * and ** indicate differences at the 0.05 and 0.01 probability levels, respectively. (**A**) Sowing stage soil water storage (**B**) Jointing stage soil water storage (**C**) Anthesis stages soil water storage (**D**) Maturity stages soil water storage.

### 3.2. Correlation between Yield Formation and Field Water Consumption

During growth, the correlation between water consumption and yield formation was different at each yield level (Figure 5). At a low yield level, yield increased with increasing soil water consumption during each growth stage, although the differences were not significant (Figure 5A–C). Yield increased with field water consumption during the jointing to anthesis stages at the intermediate yield level, as shown in Figure 5B. On other hand, at the high yield level, yield increased with water consumption during the anthesis and maturity stages (Figure 5A–C). These results indicate that higher field water consumption during late growth stages is essential to high yield.

### 3.3. Water Use Efficiency (WUE) and Yield Components

During the research period from 2009–2017, the lowest yield was recorded under the DS treatment in 2012–2013, while the highest yield was recorded under furrow sowing (FS) in 2015–2016, as shown in Table 2. Moreover, yield composition was different at the different yield levels. In 2012–2013, at a low yield level, the 1000-grain weight and grain number per spike were highest under the WS treatment, while the lowest yield was noted under drill sowing, with values of 300.25 × 104 ha$^{-1}$ and 2.14 kg·ha$^{-1}$ recorded for grain

yield and tiller number, respectively. Meanwhile, at the intermediate yield level, the yield, number of tillers, and grain number per ear were highest under DS in 2016–2017. At the lowest yield recorded in 2015–2016, the tiller number was also lowest. FS and WS treatments increased the number of tillers, grain number per spike, and 1000-grain weight, thereby increasing grain yield by 26.5%, and 24.5%, respectively, compared to DS. At the low yield level, the average field water consumption, WUE, and PUE were 334.7 mm, 7.8 t·ha$^{-1}$, and 7.6 t·ha$^{-1}$·mm$^{-1}$, respectively, while the water consumption was highest in the year with the highest yield, and the WUE was also relatively high. In addition, compared with DS, FS and WS effectively improved the WUE by 11.7% and 11.9% and the PUE by 26.7% and 24.2%, respectively, in the same year.

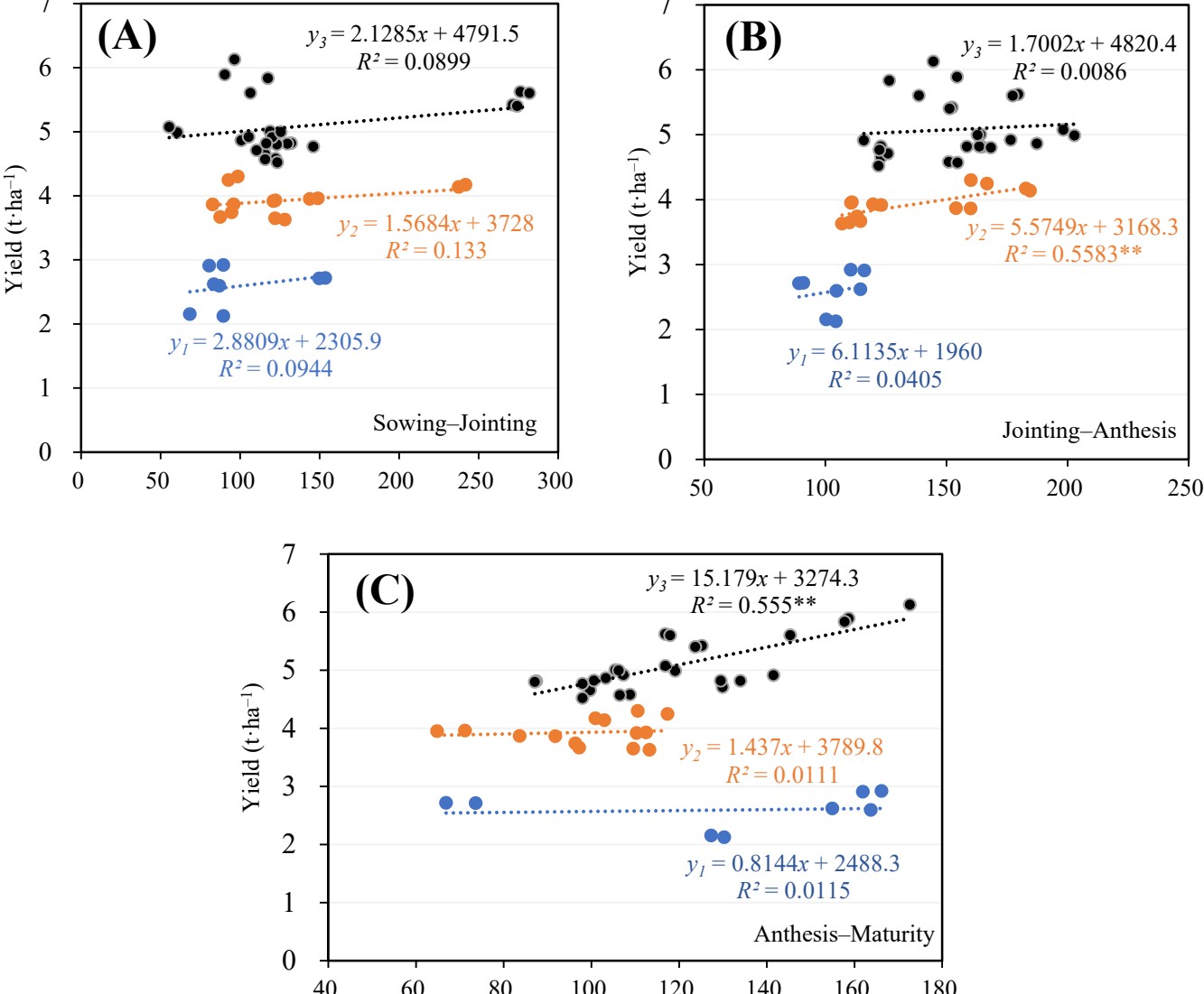

**Figure 5.** Correspondence between field water intake at different growth stages and yield using different sowing methods ($y_1$ = DS, drill sowing; $y_2$ = FS, furrow sowing; $y_3$ = WS, wide-space sowing); (**A**) Jointing stage soil water storage, (**B**) Jointing stage- anthesis stages soil water storage, (**C**) Anthesis stages-maturity stages soil water storage; * and ** indicate differences at the 0.05 and 0.01 possibility levels, respectively.

**Table 2.** Differences in yield components and WUE under DS, FS, and WS treatments.

| Sowing Methods | Tillers ($10^4$ ha$^{-1}$) | Grain Number per Spike | 1000-Grain Weight (g) | Yield (t·ha$^{-1}$) | Evapotranspiration (mm) | Water Use Efficiency (WUE; kg·h$^{-1}$·mm$^{-1}$) | Precipitation Use Efficiency (PUE; kg·h$^{-1}$·mm$^{-1}$) |
|---|---|---|---|---|---|---|---|
| 2009–2010 DS | 407.71 [a] | 20.38 [c] | 36.14 [c] | 2714.96 [b] | 311.98 [c] | 8.70 [a] | 8.10 [b] |
| 2012–2013 DS | 300.25 [d] | 20.37 [c] | 36.46 [c] | 2140.25 [d] | 310.17 [c] | 6.90 [d] | 6.24 [d] |
| 2012–2013 FS | 341.50 [c] | 22.29 [b] | 38.81 [b] | 2608.30 [c] | 354.10 [b] | 7.37 [c] | 7.61 [c] |
| 2012–2013 WS | 350.25 [b] | 23.17 [a] | 40.67 [a] | 2915.32 [a] | 362.43 [a] | 8.04 [b] | 8.50 [a] |
| Mean | 349.93 | 21.55 | 38.02 | 2594.71 | 334.67 | 7.75 | 7.61 |
| 2009–2010 FS | 427.18 [c] | 21.70 [f] | 39.04 [c] | 3639.82 [f] | 344.88 [d] | 10.55 [f] | 10.87 [b] |
| 2009–2010 WS | 453.72 [b] | 23.78 [e] | 42.08 [a] | 3923.57 [c] | 354.37 [c] | 11.07 [e] | 11.71 [a] |
| 2010–2011 DS | 401.04 [e] | 26.22 [c] | 40.51 [b] | 3705.67 [e] | 301.65 [g] | 12.28 [a] | 6.93 [f] |
| 2011–2012 DS | 485.50 [a] | 24.33 [d] | 35.44 [d] | 4155.60 [b] | 525.20 [a] | 7.91 [g] | 6.17 [g] |
| 2013–2014 DS | 386.65 [f] | 27.55 [b] | 39.12 [c] | 3866.73 [d] | 334.05 [e] | 11.58 [c] | 8.15 [d] |
| 2014–2015 DS | 417.00 [d] | 27.48 [b] | 39.14 [c] | 3956.22 [c] | 325.22 [f] | 12.16 [b] | 7.66 [e] |
| 2016–2017 DS | 452.12 [b] | 33.36 [a] | 35.66 [d] | 4274.00 [a] | 373.02 [b] | 11.46 [d] | 10.52 [c] |
| Mean | 431.89 | 26.35 | 38.71 | 3931.66 | 365.48 | 11.00 | 8.86 |
| 2010–2011 FS | 446.58 [k] | 28.24 [g] | 40.59 [c,d] | 4588.15 [h] | 340.81 [j] | 13.46 [c] | 8.58 [i] |
| 2010–2011 WS | 481.08 [h] | 28.38 [f,g] | 42.58 [a] | 4794.56 [f] | 361.01 [i] | 13.28 [c] | 8.97 [h] |
| 2011–2012 WS | 603.00 [b] | 26.56 [h] | 37.15 [f] | 5412.04 [d] | 549.04 [b] | 9.86 [h] | 8.04 [k] |
| 2011–2012 FS | 616.50 [a] | 26.74 [h] | 38.63 [e] | 5612.45 [c] | 575.02 [a] | 9.76 [h] | 8.34 [j] |
| 2013–2014 FS | 454.41 [j] | 28.31 [f,g] | 41.04 [b,c] | 4575.40 [h] | 379.48 [f] | 12.06 [f] | 9.65 [f] |
| 2013–2014 WS | 466.00 [i] | 29.63 [e] | 41.55 [b] | 4818.74 [f,g] | 409.82 [c] | 11.76 [g] | 10.16 [e] |
| 2014–2015 FS | 488.33 [f,g] | 28.79 [f] | 40.30 [d] | 4806.55 [f,g] | 380.16 [f] | 12.64 [e] | 9.30 [g] |
| 2014–2015 WS | 522.98 [c] | 29.72 [e] | 41.01 [b,c] | 4999.96 [e] | 391.54 [e] | 12.77 [d,e] | 9.68 [f] |
| 2015–2016 DS | 425.75 [l] | 34.78 [d] | 39.06 [e] | 4812.00 [f,g] | 371.90 [h] | 12.94 [d] | 12.44 [c] |
| 2015–2016 WS | 484.50 [g,h] | 36.23 [b] | 39.11 [e] | 5719.08 [b] | 396.09 [d] | 14.44 [b] | 14.79 [b] |
| 2015–2016 FS | 493.25 [e,f] | 37.80 [a] | 41.26 [b] | 6009.75 [a] | 408.60 [c] | 14.71 [a] | 15.54 [a] |
| 2016–2017 WS | 496.25 [e] | 35.57 [c] | 33.12 [h] | 4892.00 [f] | 390.33 [e] | 12.53 [e] | 12.04 [d] |
| 2016–2017 FS | 503.36 [d] | 35.54 [c] | 34.21 [g] | 5032.00 [e] | 376.52 [g] | 13.36 [c] | 12.38 [c] |
| Mean | 498.61 | 31.25 | 39.20 | 5082.51 | 410.02 | 12.58 | 10.76 |
| | | | | ANOVA | | | |
| Sowing (S) | <0.001 | <0.001 | <0.001 | <0.001 | <0.001 | <0.001 | <0.001 |
| Year (Y) | <0.001 | <0.001 | <0.001 | <0.001 | <0.001 | <0.001 | <0.001 |
| S × Y | <0.001 | <0.001 | <0.001 | <0.001 | <0.001 | <0.001 | <0.001 |

Note: DS = drill sowing, FS = furrow Sowing, WS = wide-space sowing. Significant differences between different yield levels are indicated by different letters in the same treatment ($p < 0.05$).

### 3.4. Correlation Analysis of Yield Components and Contribution of Water Sources

The contribution of the different yield components to yield varied with yield level (Table 3). Thus, at a low yield level, the number of tillers and 1000-grain weight were positively correlated with yield. Meanwhile, at the intermediate yield level, the 1000-grain weight was negatively associated with yield, while the number of tillers and number of grains per spike were the key mechanisms for increasing yield. The association between the 1000-grain weight and yield was nonsignificant, whereas the yield was significantly improved by the number of tillers.

**Table 3.** Correlation between yield and components.

| Sowing Methods | Tillers | Number per Spike | 1000-Grain Weight | Simulation Equation |
|---|---|---|---|---|
| DS | 0.676 ** | 0.661 * | 0.634 * | $Y = 5.694 \times Y1 + 111.949 \times Y3 - 3653.974$, $R^2 = 0.999$ |
| FS | 0.626 ** | 0.641 ** | −0.700 ** | $Y = 4.558 \times Y1 + 42.942 \times Y2 + 831.857$, $R^2 = 0.999$ |
| WS | 0.540 ** | 0.375 * | −0.088 | $Y = 8.836 \times Y1 + 111.52 \times Y2 + 93.9 \times Y3 - 6489.48$, $R^2 = 0.999$ |

Note: DS = drill sowing, FS = furrow sowing, WS = wide-space sowing. * $p < 0.05$. ** $p < 0.01$.

At the low yield level, fallow precipitation and the jointing to anthesis stages were positively correlated with the number of tillers; however, this correlation was negative during the anthesis to maturity stages (Table 4). The number of grains per spike and the 1000-grain weight were negatively correlated with precipitation. Soil water consumption during the sowing to jointing stages was positively correlated with the number of tillers. The grain number per spike and the 1000-grain weight were positively correlated with

soil water consumption during the jointing to anthesis stages and during the anthesis to maturity stages. At the intermediate yield level, precipitation during the sowing to jointing stages was positively correlated with the number of tillers; however, this correlation was negative during the jointing to anthesis stages and during the anthesis to maturity stages. Precipitation during the sowing to jointing stages was negatively correlated with the grain number per spike. Lastly, the correlation between precipitation and the number of tillers at the high yield level was similar to that detected at the low yield level. Furthermore, fallow precipitation and precipitation during the sowing to jointing stages were negatively correlated with the grain number per spike, whereas this correlation was positive during the jointing to anthesis stages and during the anthesis to maturity stages. On the other hand, soil water consumption during the sowing to jointing stages and during the jointing to anthesis stages was positively correlated with the number of tillers, whereas the correlation with water consumption was negative during the anthesis to maturity stages.

The equations in Table 4 show that soil water intake from anthesis to maturity mostly influenced the number of grains per spike and 1000-grain weight under drill sowing. Furthermore, the number of tillers was positively affected by soil water consumption from jointing to maturity, the grain number per ear was affected by soil water consumption from anthesis to maturity, and the 1000-grain weight was affected by precipitation from seeding to jointing and by soil water consumption from anthesis to maturity. Lastly, the number of tillers was positively affected by fallow precipitation from seeding to anthesis at a high yield level, the grain number per spike was affected by water consumption from seeding to anthesis and by precipitation from jointing to maturity, and the 1000-grain weight was affected by fallow precipitation and precipitation during each growth stage.

**Table 4.** Correlation between yield components and water source contribution.

| Sowing Methods | Yield Composition | Fallow Precipitation | Soil Water Consumption Sowing–Jointing | Precipitation Sowing–Jointing | Soil Water Consumption Jointing–Anthesis | Precipitation Jointing–Anthesis | Soil Water Consumption Anthesis–Maturity | Precipitation of Anthesis–Maturity | Simulation Equation |
|---|---|---|---|---|---|---|---|---|---|
| DS | Tillers | 0.869 ** | 0.951 ** | 0.869 ** | −0.698 ** | 0.869 ** | −0.199 | −0.869 ** | $Y1 = 1.345 \times X6 - 2.108 \times X7 + 489.556, R^2 = 0.999$ |
| | Grain number per ear | −0.551 * | −0.338 | −0.551 * | 0.765 ** | −0.551 * | 0.949 ** | 0.551 * | $Y2 = 0.064 \times X6 + 19.014, R^2 = 0.999$ |
| | 1000-grain weight | −0.585 * | −0.370 | −0.585 * | 0.779 ** | −0.585 * | 0.944 ** | 0.585 * | $Y3 = 0.097 \times X6 + 34.21, R^2 = 0.999$ |
| FS | Tillers | 0.012 | −0.033 | 0.812 ** | 0.665 ** | −0.611 ** | 0.939 ** | −0.483 * | $Y1 = 0.267 \times X4 + 1.513 \times X6 + 326.621, R^2 = 0.999$ |
| | Grain number per ear | −0.112 | −0.785 ** | 0.368 | 0.242 | 0.120 | −0.167 | 0.069 | $Y2 = 0.01 \times X1 - 0.097 \times X2 - 0.1 \times X6 - 0.131 \times X7 + 38.279, R^2 = 0.999$ |
| | 1000-grain weight | −0.212 | 0.360 | −0.869 ** | −0.730 ** | 0.259 | −0.470 * | 0.356 | $Y3 = -0.058 \times X3 + 0.051 \times X6 + 41.543, R^2 = 0.98$ |
| WS | Tillers | 0.345 * | 0.630 ** | 0.819 ** | 0.524 ** | 0.629 ** | −0.482 ** | −0.559 ** | $Y1 = 0.375 \times X1 + 0.97 \times X3 + 0.732 \times X5 + 355.131, R^2 = 0.999$ |
| | Grain number per ear | −0.872 ** | −0.099 | −0.478 ** | −0.949 ** | 0.311 * | 0.253 | 0.695 ** | $Y2 = 0.053 \times X3 - 0.039 \times X4 - 0.021 \times X5 - 0.077 \times X6 + 44.642, R^2 = 0.99$ |
| | 1000-grain weight | 0.605 ** | −0.708 ** | −0.211 | 0.269 * | −0.822 ** | 0.451 ** | 0.160 | $Y3 = 0.017 \times X1 + 0.061 \times X3 - 0.051 \times X5 + 0.051 \times X7 + 37.26, R^2 = 0.999$ |

Note: DS = drill sowing, FS = furrow sowing, WS = wide-space sowing. * $p < 0.05$. ** $p < 0.01$.

## 4. Discussion

### 4.1. Wheat Grain Yield and Yield Components

Precipitation is the only source of water in arid and semiarid areas; therefore, it is the main preventive factor for the production of winter wheat [29]. Field water consumption, precipitation use efficiency, and water use efficiency were affected by the tillage treatment, thereby affecting the winter wheat yield [30]. In addition, wheat yield was significantly correlated with soil water status at numerous developmental stages from sowing to maturity [31]. In a previous study, it was reported that soil water storage from jointing to maturity was the key factor for increasing winter wheat yield in the Loess Plateau region [32], with the main stages for the water demand of winter wheat being sowing, jointing, and anthesis [33].

Soil moisture during the jointing and heading stages is particularly important in determining yield formation. The correlation between yield and soil water storage during each growth stage was different, not only related to regional differences but also to yield level [34]. In a previous study, when yield was lower than 3.00 t·ha$^{-1}$, it was more strongly related to soil water storage at sowing, jointing, and anthesis [35]. When yield reached between 3.10 and 4.51 t·ha$^{-1}$, it was more related to soil water storage at jointing, whereas, when it reached over 4.50 kg·ha$^{-1}$, it was more related to soil water storage at jointing, maturity, and anthesis [36]. In the fallow period, tillage improved the soil water storage and field evapotranspiration, which was conducive to the improvement of yield [37]. Optimizing the spike number per hectare is a key method to maximize yield in most cereal crops because it can increase plant vigor and, hence, plant grain yield [38].

Both the number of tillers and the yield were positively correlated at different yield levels, indicating that a larger number of tillers may guarantee a higher yield from winter wheat. These results are consistent with previous studies [39]. However, the grain number per ear and 1000-grain mass were correlated with yield at different levels of yield. Thus, for example, [39] reported that, at a low yield level (less than 7.50 t·ha$^{-1}$), yield was positively correlated to grain number per spike ear but negatively correlated with 1000-grain mass, whereas, at a high yield level (i.e., greater than 7.50 t·ha$^{-1}$), yield was correlated with grain number per ear, but not with 1000-grain weight. In the present study, a significant relationship was found between yield and tiller number. However, when the yield was lower than 3.00 t·ha$^{-1}$, it was correlated with 1000-grain weight, whereas, when the yield was 3.00 and 4.50 t·ha$^{-1}$, it was significantly and negatively correlated with 1000-grain weight. In addition to the number of tillers, at low and intermediate yield levels, the 1000-grain mass and the number of grains per spike were the key yield components responsible for increasing crop yield. Similarly, at a high yield level, higher values of grain number per spike, 1000-grain weight, and number of tillers were the key to high crop yield.

### 4.2. Wheat Yield Formation and Water

The key yield components responsible for the formation of yield are well known to be affected by soil moisture during each growth stage and to influence each other [40]. The early growth stage is conducive to improving the spike number, while the latter growth stage is important for the spike number and 1000-grain weight [41]. The number of tillers was reported to be more closely related to water content at the early stage of growth at different yield levels, and the number of grains per ear and 1000-grain weight were more closely related to growth stage, although the specific correlation varied, especially the relative contribution to the formation of the different yield components [42]. Thus, at low yield levels, the key to improving tiller number and 1000-grain weight was soil water consumption during the period from anthesis to maturity [43]. At the intermediate yield level tested here, tiller number was affected by soil water consumption during jointing, and the effect was positive; the number of grains per spike was positively affected by water consumption during the period from anthesis to maturity [44]. In this study, the fallow period ranged from the last 10 days of June to the last 10 days of September, the

sowing–jointing stage) ranged from the first 10 days of October to the first 10 days of April in the following year, the jointing–anthesis stage ranged from the middle 10 days of April to the first 10 days of May, and the anthesis–maturity stage ranged from the middle 10 days of May to the middle 10 days of June.

Soil erosion has disastrous consequences on local agricultural creation and socioeconomic improvement, thereby affecting people's lives and property, and posing a significant threat to safety. Loess erosion is a main environmental topic that has been addressed in many studies [45–47]. The results of previous studies revealed large differences in soil temperature and moisture across tillage and sowing treatments in wheat [48,49]. In agriculture systems, the method of sowing is an important factor governing the soil microclimate [50,51]. Unlike tillage systems, crop residues are not incorporated in sowing systems [52,53]. The amount of soil water stored at sowing can be used as a guide when applying the basal amount of N. Additional N fertilization as top dressing can be applied when rainfall is higher than expected in the growth season [54,55]. The annual precipitation level fluctuates considerably in the Loess Plateau, as observed in this study [56].

Precipitation is also unevenly distributed within a year. Summer rainfall accounts for approximately 60% of the yearly precipitation [57,58]. The yield increase is largely because optimal sowing promotes tiller and panicle formation, leading to an increased number of spikes per unit area (Table 4). The application of optimal rates of N in years with different precipitation levels can also reduce production cost and environmental pollution [59,60]. In turn, the 1000-grain weight was found to be affected by precipitation from sowing to jointing and by soil water consumption from anthesis to maturity. Lastly, at the high yield level, the number of tillers was positively affected by fallow precipitation during the sowing and anthesis periods, the number of grains per ear was affected by water consumption during the jointing–maturity stages and by precipitation from sowing to anthesis, and the 1000-grain weight was affected by fallow precipitation during each growth stage.

*4.3. Water Impact on Wheat Yield*

This study showed that sowing method had no significant effect on the grain protein content. Compared to drill sowing, the protein yield of wheat could be significantly increased by wide-space sowing, and the soil moisture could be significantly increased by furrow sowing. Furthermore, the regulation ability of wide-space sowing was higher than that of furrow sowing. The results showed that the difference in protein yield was mainly caused by yield, in contrast to the results in Tai'an, Shadong province [61], where, compared to drill sowing, wide-space sowing could reduce wheat grain protein content and increase protein yield. This may be due to the differences in regional climate and soil type or may be related to wheat genotypes, which need to be verified by years of research. Analysis showed that nitrogen fertilizer could significantly increase grain protein content and yield, and its regulation ability increased with the increase in nitrogen fertilizer. This was consistent with previous studies showing that nitrogen application increased the nitrogen content in grains [62,63], thus increasing the protein content. It was also shown that the sowing method and nitrogen application rate had independent effects on nitrogen accumulation in dryland wheat [64]. This may be due to the different response of grain protein content and yield to the seeding method and nitrogen application rate; thus, further research is needed.

This study showed that the contribution rate of pre-flowering translocation to grains was more than 75%. Compared to drill sowing, the nitrogen accumulation, transshipment volume, and N harvest index of wheat plants were significantly increased by wide-space sowing and trenching tillage sowing, whereas the contribution rate of post-flowering accumulation to grains was significantly decreased by wide-space sowing and trenching tillage sowing, along with a higher regulation ability than trenching tillage sowing. This is consistent with previous studies. The large population [64] produced by wide-space sowing and double-row sowing is accompanied by an increase in plant nitrogen accumulation [65], while the premature aging phenomenon [66,67] results in accelerated filling, high

pre-flowering transshipment volume, transshipment rate [68], and eventually high grain nitrogen content and harvest index [69,70]. The agent quantity of furrow sowing with the buffer effect of temperature [71] is advantageous to plant nitrogen accumulation and delays the grouting by 5–6 months at high temperatures [72]. Additionally, it increases the grain nitrogen content and harvest index [73]; however, of the land utilization rate is low, and the nitrogen accumulation and transportation are lower than under wide refined sowing. In conventional single-row seeding, the lack of seedlings and ridging at the early stage [74] reduces the wheat population, resulting in low nitrogen accumulation in plants and weak resistance to the external environment in smaller groups at the later stages [75]. A high temperature at the filling stage further reduces the transport of nitrogen to grains, resulting in a lower nitrogen content and harvest index in grains. Experimental results in the Loess Plateau showed that soil water storage before sowing was significantly and positively correlated with wheat yield in dryland. In the Weibei region of Shaanxi province and the Jinnan region of Gansu province, soil water storage before sowing showed a significant linear positive correlation with wheat grain yield, especially in dry years. The distribution of precipitation is closely related to wheat yield. If precipitation is insufficient in the early stages of the critical wheat growth period and a soil water deficit occurs, the growth and development of wheat will be significantly affected, resulting in a reduction in yield.

## 5. Conclusions

It can be concluded from the present study that, compared to the drill sowing method, furrow sowing and wide-space sowing were influenced by field evapotranspiration within the same year. At a low yield level, the average field water consumption, WUE, and PUE were highest in the year with the highest yield. Wide-space sowing in the fallow period improved the precipitation use efficiency, while yield components that were negatively affected by precipitation were also improved. Wide-space sowing was mainly responsible for a reduction in 1000-grain weight and grain number per spike. Therefore, in high-yield years, fallow cultivation can help adjust the relationship among the components, promote a reasonable distribution, and improve yield.

**Author Contributions:** Conceptualization, M.S.; methodology, H.N., M.S. and Z.G.; software, M.S.; validation, M.S. and H.N.; formal analysis, W.L.; investigation, M.S.; resources, H.N.; data curation, H.N.; writing, H.N., M.S., W.L. and Z.G. All authors read and agreed to the published version of the manuscript.

**Funding:** The authors are thankful to 'Modern Agriculture Industry Technology System Construction' (No. CARS-3124), the National Key Research and Development Program of China (No. 2018YFD020040105), the Sanjin Scholar Support Special Funds Projects, the National Natural Sci-ence Foundation of China (No. 31771727), and the '1331' Engineering Key Innovation Cultiva-tion Team Organic Dry Cultivation and Cultivation Physiology Innovation Team (No. SXYBKY201733).

**Institutional Review Board Statement:** Not applicable.

**Informed Consent Statement:** Not applicable.

**Data Availability Statement:** The data presented in this study is available on request from the corresponding author.

**Acknowledgments:** The authors are thankful to 'Modern Agriculture Industry Technology System Construction' (No. CARS-3124), the National Key Research and Development Program of China (No. 2018YFD020040105), the Sanjin Scholar Support Special Funds Projects, the National Natural Science Foundation of China (No. 31771727), and the '1331' Engineering Key Innovation Cultivation Team Organic Dry Cultivation and Cultivation Physiology Innovation Team (No. SXYBKY201733) for financial support of this study.

**Conflicts of Interest:** The authors declare no conflict of interest.

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
