# Peer review of "Effect of Different Sowing Methods on Water Use Efficiency and Grain Yield of Wheat in the Loess Plateau, China"

_water, doi:10.3390/w14040577_

Round 1
Reviewer 1 Report
The manuscript entitled "Different sowing methods between yield formation soil, and water use efficiency in wheat at yield levels on the Loess Plateau, China" presents an interesting experimental study conducted on the water use during the winter wheat growing season in the Loess Plateau from China. However, the results of the study have been presented with limited discussions and other issues must be addressed. The paper needs minor revisions before it is processed further, some comments follow:
Introduction section
The introduction section can be improved. The citations have been introduced in bulk form "[9-11]", [12-14], [15-17] etc. and not distributed in the text in accordance with the affirmations that must be supported. Please introduce citation at a specific position to assure a clear correspondence between the affirmations from the introduction section and the previous publication.
Materials and Methods Sections
Figure 2- please improve the scientific level of the figure. Please highlight the area of interest for the reader. Also, please translate the panels from the right column to English.
Conclusions Sections
The conclusions are really weak, please extend them, why did you not follow the main recommendations by Academia of giving the conclusions of the study by points with highlights.
Reviewer 2 Report
This manuscript is a good start to understanding which conservation tillage establishment of winter wheat is best for wheat productivity and water use in rain-fed production systems in the Loess Plateau, China. However, the co-authors need to make improvements to the manuscript in order for this to be suitable for publication in MDPI Water. I would be willing to review a revised version of this manuscript after co-authors have made the following FOUR general edits in addition to line-number specific edits:
1) Please expand the Introduction to provide more context in terms of why wheat production is important globally and why specifically this region in China is important (e.g., percent of global production, etc.) specifically. Also, wheat production regions around the globe may be adversely impacted over the next century due to climate change. Is this region in China projected to be sensitive to such projected changes? Also, is water availability from rivers, groundwater, and reservoirs limited in this region of China? If so, then this would further emphasize why your research is important but this needs to be clearly outlined and explained as part of the expansion of the Introduction section.
2) The Discussion section typically has two sections:
a) Expand on the contrasts and linkages between your results and results of prior literature. You have done this but the three subsections can be made into sub-sub-sections under this newly created sub-section of “1. Comparison to Prior Studies” and then have the following sub-sub-sections under this:
4.1.1. Water impacting wheat yield
4.1.2. Wheat grain yield and yield components
4.1.3. Wheat yield formation and water
This leaves a newly created sub-section “4.2. Research Implications” to finish the Discussion section and write about b) below.
b) Discuss implications of your research results. This needs to be added. For example, your results like prior literature demonstrate that direct seeding takes a yield penalty. Why is this? Are there any documented benefits environmentally that favor direct seeding versus the other two reduced tillage systems you evaluated? For example, the literature shows that direct seeding disrupts the soil less and results in less CO2 efflux from the soil than more intensive soil tillage, especially when combined with greater soil moisture and higher temperatures. There are several studies from northern China on this. Are these potential benefits worth taking a yield penalty for? Will these potential tradeoffs change given projected changes in climate to your study region in China?
3) Please add a couple of sentences to the Conclusions section on how your research can be improved in general in the future.
4) Please cite and list more references to get to 50 or more. By adding to the Introduction and Discussion sections, the number of cited sources will increase.
Specific Line Number of Manuscript Figure/Table comments (Please note that “…” symbolizes writing that is not changed on either side of the requested edit):
L2-4 – Improve the title to interest the reader in reading your manuscript…for example what is “yield formation soil”?
L107 – Do not indent this line
L168 & L171 – Superscript so “ha-1”
L179 – Superscript so “kg ha-1 mm-1” for both
L197 – Add blank row below this line (footnote for Table 4)
L310-386 – The format of the cited references is not correct. The year of journal articles needs to be in bold. You need to add the volume (issue) numbers which are in italics. The journal names also need to be in italics in addition to being abbreviated. Please refer to the MDPI Water template for this online.
Please note that in general the English is good but I do a final round of typographical edits once the next manuscript draft is submitted with the four substantive edits are addressed above. Thanks!
Round 2
Reviewer 2 Report
This manuscript is a good start to understanding which conservation tillage establishment of winter wheat is best for wheat productivity and water use in rain-fed production systems in the Loess Plateau, China. The manuscript is close to being acceptable for publication in MDPI Water after the co-authors make minor improvements and clarifications to the manuscript. I would like to review the manuscript after these edits are made. The co-authors need to make the following SIX general edits in addition to line-number specific edits:
1) The current paragraph format is not indented and with a blank line separating paragraphs so you will have to change that to indenting paragraphs with NO blank line separating paragraphs
2) You need to break up the writing into shorter paragraphs that are logically organized by topic and that flow from one to the next, particularly in the entire Discussion section. A paragraph by definition is at least 3 sentences. You have writing that only has 1 or 2 sentences.
3) On L92-100 for Figure 1, please clarify in the title what the colors represent and include a legend. I assume it is precipitation (or is it elevation?)? but it is pointless to use colors if the reader does not know what the interval ranges are. Please keep the colors as the figure looks great but you need to make the figure more clear in terms of what it represents.
4) Figure 5 yield y-axis labels are not correct. Should be metric tons (t) and not kilograms. You do not yield only 5 kg of wheat from a hectare. Please double check units are correctly labeled everywhere else in the manuscript. Also, please for all 3 graphs add error bars to data points.
5) In the Discussion for L387-399, the writing is just stating results again and not contrasting to the results from prior literature cited. Please draw the contrasts like you do in the previous sub-section. The added writing on L399-417 needs to be better organized and integrated with the original writing. Please consider the following big picture summary of what I get from your results:
a) Wide-Space (WS) sowing gets higher yields relative to Furrow Sowing (FS) and traditional grain Drill Seeding (DS) and this appears to be significant.
b) WS is the only system responsive to water/precipitation.
c) Loess Plateau is rainfed so unlike irrigated systems, taking advantage of the yield response for WS is challenging but perhaps worth doing
Which systems is more prevalent in the region? Your results suggest the entire region should adopt WS and get rid of FS and DS. Are there reasons why this may not or should not be done? These questions you need to write about after creating an outline. You added a couple sentences here on erosion with no context. Does the WS since there is more bare soil more subject to the wind erosion or potential water-borne erosion prevalent in the Loess Plateau? If so, then there are environmental challenges to WS relative to the other two systems and a trade-off of yield versus environmental impact.
6) Please add a couple of sentences to the Conclusions section on how your research can be improved upon in general in the future. You have added more writing on your results but typically Conclusions end on broader picture description on how to direct future work.
Specific Line Number of Manuscript Figure/Table comments (Please note that “…” symbolizes writing that is not changed on either side of the requested edit):
L22 – Add “wheat” to keywords
L42 – Capitalize the word “Sowing” in 3 places since the abbreviation letter “S” is capitalized
L53 – Change to “…of nitrogen (N) fertilizers can have several negative”
L54 – Do NOT capitalize “study”
L55 – Change to “of controlled-release nitrogen fertilizers at sowing, the…”
L60-61 – Change to “rate, the investment in N application should be determined to optimize economic return [21].”
L135 – Unbold and change to “2.1.”
L166 – Do not bold and no not italicize and change to “2.2”
L167 – Keep in italics but do NOT bold
L170 – Change to “…intervals as described in [28].”
L171 – Change to “…obtained using equations 1 and 2, respectively:”
L172-173 – Do NOT bold
L178 – Change to “…calculated using equations 3, 4, and 5:”
L179-181 – Do NOT bold
L189 – Add a 1st sentence to start the paragraph informing the reader that the water loss is not only from soil but through plants as well
L190 – Change to “…growth periods, The sum”
L194 – Do NOT bold
L198 – Change to “shelling them mechanically. Then the grain…”
L199 – Do NOT bold and do NOT italicize
L205 – Change to “t-test” since unlike “F-test” this is not capitalized
L207 – Do NOT italicize
L208 – Do NOT bold and do NOT italicize
L215 – Change to “…water storage at the jointing”
L217 – Change to “…and maturity. This trend was similar…”
L218 – Change start of sentence to “Our results indicate…”
L249 – Do NOT bold and do NOT italicize
L252-253 – Change to “…were not significant (Figure 4A-C).”
L255 – Change to “level as shown in Figure 5B.”
L257 – Change to “(Figure 5A-C).”
L259-282 – For all 3 graphs, the x-axis is not labeled…you do not need to label for all but just the bottom. So the water consumption is measured in mm? Also on the first graph, the number 300 is cut off.
L386 – Do NOT bold and do NOT italicize
L399 – Start new paragraph with new writing
L403 – Add space after period
L421 – Change to “stages”
Round 3
Reviewer 2 Report
This manuscript is a good start to understanding which conservation tillage establishment of winter wheat is best for wheat productivity and water use in rain-fed production systems in the Loess Plateau, China. The manuscript is close to being acceptable for publication in MDPI Water after the co-authors make the minor improvements and clarifications to the manuscript that I outlined in my previous review (many of these edits were NOT made). I would like to review the manuscript after these edits are made. The co-authors need to make the following SIX general edits in addition to line-number specific edits:
1) On L92-93 for Figure 1, please clarify in the title what the colors represent and include a legend. I assume it is precipitation (or is it elevation?)? This correction was not done.
2) For some reason you delete the “(#)” at the end of each equation line and then proceeded to bold all equations. Do not bold equations and please add back in “(#)” making sure these are right justified all the way to the right and are aligned with other “(#)” below and above.
3) You need to break up long tracks of writing into paragraphs with at least 2 sentences. This makes it a lot easier for the reader to understand. Please do this on L26-51, L52-79, L264-298, and L300-335.
4) Some of the edits you have made highlighted in yellow have made the English writing worse. Please have someone in the English department read the manuscript, edit for English, and then use these changes.
5) The requested added discussion of your results was not done. You need to critically think about this and write this as answering the “SO WHAT?” question regarding your research results is an essential part of journal articles. Please add another sub-section to the Discussion section “4.3.” that addressed the following discussion on the SYSTEMS you evaluated:
Your Results:
a) Wide-Space (WS) sowing gets higher yields relative to Furrow Sowing (FS) and traditional grain Drill Seeding (DS) and this appears to be significant.
b) WS is the only system responsive to water/precipitation.
c) Loess Plateau is rainfed so unlike irrigated systems, taking advantage of the yield response for WS is challenging but perhaps worth doing
Discussion on these Results (added as new sub-section 4.3.):
Which systems is more prevalent in the region? Your results suggest the entire region should adopt WS and get rid of FS and DS. Are there reasons why this may not or should not be done? These questions you need to write about after creating an outline. You added a couple sentences here on erosion with no context. Does the WS since there is more bare soil more subject to the wind erosion or potential water-borne erosion prevalent in the Loess Plateau? If so, then there are environmental challenges to WS relative to the other two systems and a trade-off of yield versus environmental impact.
6) Please add a couple of sentences to the Conclusions section on how your research can be improved upon in general in the future. You have added more writing on your results but typically Conclusions end on broader picture description on how to direct future work.
Specific Line Number of Manuscript Figure/Table comments (Please note that “…” symbolizes writing that is not changed on either side of the requested edit and I do this so I do not need to write out the rest of the line(s)):
L19 – Delete the first word “were” (this is not correct English)
L17-22 – Break up into two sentences to increase clarity
L23 – Add “wheat” to keywords in alphabetical order with respect to other key words
L32 – Do NOT capitalize so change to “winter”
L55 – Change to “…environment [18]. A previous study showed that use of”
L56 – Change to “of controlled-release nitrogen fertilizers at sowing increased crop yield…”
L128 – Change to “2.2.”
L129 – Change to “2.2.1.”
L144 – Line should read “given cropping season calculated using equations 3, 4, and 5:” WITHOUT the quotation marks (Please correct any error interpreting my edits like this anywhere else in the manuscript).
L161 – This is a perfect example of your edit of adding “apiece” makes the sentence not proper English…the line should read:
Fifty plants per plot were randomly sampled at maturity from the inner rows
L200 – Why is the “5A” in blue and not in black?
L208 – Why is the “2” in blue and not in black?
L236 – Why is the “4” in blue and not in black?
L202-203 – For all 3 graphs, the x-axis is not labeled…you do not need to label for all but just the bottom. So the water consumption is measured in mm? Also on the first graph, the number 300 is cut off.
Round 4
Reviewer 2 Report
This manuscript is a good start to understanding which conservation tillage establishment of winter wheat is best for wheat productivity and water use in rain-fed production systems in the Loess Plateau, China. The manuscript is close to being acceptable for publication in MDPI Water after the co-authors make the minor improvements and clarifications to the manuscript that I outlined in my previous review (some of these edits were NOT made). I would like to review the manuscript after these edits are made. The co-authors need to make the following FOUR general edits in addition to line-number specific edits:
1) On L91-94 for Figure 1, it is great that you have clarified in your response to reviewer comments that the colors represent precipitation. You still however need to include a legend. The legend will show which precipitation ranges correspond to which colors and this is typically an inset box which I would suggest you put in the bottom left corner of the map. Please add the “* Experiment site” to this box since this is part of the legend that tells the reader what the “*” on the map represents as well as what the colors mean.
2) The edits to the English writing requested were not done. Please take the time to have someone in the English department edit the manuscript for proper English writing. This should take at least a week to make these and other corrections. If this is not possible, I would be willing to edit the revision but would need the most recent version in Word and I will not be able to proceed with this until February 1, 2022 as I have a pressing project due date at the very end of January 2022. Thanks!
3) The requested added discussion of your results was not done. You need to critically think about this and write this as answering the “SO WHAT?” question regarding your research results is an essential part of journal articles:
Discussion on these Results (what you have currently written as new sub-section 4.3. is just primarily re-statements of your results which is not discussion):
First off, change the sub-section title to “4.3. Implications of research results” so it is clear what you are writing about. Which systems is more prevalent in the region? Your results suggest the entire region should adopt WS and get rid of FS and DS. Are there reasons why this may not or should not be done? These questions you need to write about after creating an outline. Does the WS since there is more bare soil more subject to the wind erosion or potential water-borne erosion prevalent in the Loess Plateau? If so, then there are environmental challenges to WS relative to the other two systems and a trade-off of yield versus environmental impact.
4) What you have written at the end of the Conclusion section is just a re-statement of your results. What I asked you to do is to write a simple sentence or two on how FUTURE research would improve on your work. Please add a couple of sentences to the Conclusions section on how your research can be improved upon in general in the future. Typically Conclusions end on broader picture description on how to direct future work.
Specific Line Number of Manuscript Figure/Table comments (Please note that “…” symbolizes writing that is not changed on either side of the requested edit and I do this so I do not need to write out the rest of the line(s)):
L105 – Add blank space after end of paragraph and before Figure 2
L117 – Unbold the “3).”
L148 – Change to “…was deeper than 50 m in”
L195 – For all 3 graphs above, the x-axis is not labeled…you do not need to label for all but just the bottom. So the water consumption is measured in mm?
L200 – Why is the “2” in blue and not in black? It is not a hyperlink.
L219 – Why is the “3” in blue and not in black? It is not a hyperlink.
L229 – Why is the “3” in blue and not in black? It is not a hyperlink.
L306 – Add a period to the end of the sentence
